# 3-D Cell Culture Systems in Bone Marrow Tissue and Organoid Engineering, and BM Phantoms as In Vitro Models of Hematological Cancer Therapeutics—A Review

**DOI:** 10.3390/ma13245609

**Published:** 2020-12-09

**Authors:** Dasharatham Janagama, Susanta K. Hui

**Affiliations:** Department of Radiation Oncology, Beckman Research Institute, City of Hope National Medical Center, Duarte, CA 91010, USA

**Keywords:** bone and marrow, bone marrow organoids, BM phantoms, biomimetic 3-D scaffolds, hematological cancer, tissue engineering, drug resistance

## Abstract

We review the state-of-the-art in bone and marrow tissue engineering (BMTE) and hematological cancer tissue engineering (HCTE) in light of the recent interest in bone marrow environment and pathophysiology of hematological cancers. This review focuses on engineered BM tissue and organoids as in vitro models of hematological cancer therapeutics, along with identification of BM components and their integration as synthetically engineered BM mimetic scaffolds. In addition, the review details interaction dynamics of various BM and hematologic cancer (HC) cell types in co-culture systems of engineered BM tissues/phantoms as well as their relation to drug resistance and cytotoxicity. Interaction between hematological cancer cells and their niche, and the difference with respect to the healthy niche microenvironment narrated. Future perspectives of BMTE for in vitro disease models, BM regeneration and large scale ex vivo expansion of hematopoietic and mesenchymal stem cells for transplantation and therapy are explained. We conclude by overviewing the clinical application of biomaterials in BM and HC pathophysiology and its challenges and opportunities.

## 1. Introduction

The bone and marrow (BM) are complex and vital tissues responsible for the support, production, and maintenance of hematopoietic stem cells (HSCs), which differentiate into myeloid and lymphoid progenitor cells and their lineages. Various cell types are involved to maintain the HSC niches. To restore the damaged BM tissues, efforts are made to reconstruct the BME niches in vitro mimicking in vivo conditions. Synthetic and natural biocompatible and biodegradable materials, cells, growth factors, and tissue engineering technologies are used to fabricate 3-D scaffolds to mimic the natural BM physiology and functions. Similarly, the HC environment-mimicking 3-D scaffolds are important for understanding the cancer biology and physiopathology in vitro, and for evolving better therapeutics. The xenograft animal model for cancer drug testing is widely used [1,2]. However, a non-animal model of biomimetic 3-D scaffolds not only complements the existing system in practice, but is also cost-effective and convenient [3]. Compatible tissue engineering potentially complements existing cancer models to ultimately lead to advances in cancer prevention, early detection of aggressive cancer, diagnosis, and treatment. Only sporadic and scattered literature is available on bone marrow tissue engineering (BMTE) and hematological cancer tissue engineering (HCTE). Various biomimetic scaffolds are in use for reconstruction of the BM niche. Co-cultured hematopoietic stem cells with other BM component systems model the BM niche. The organoids grow rapidly and cost effectively facilitating direct testing of a drug or combination of drugs to predict clinical response from patient. The hematological tumor organoids are proven to be more sensitive or resistant to chemotherapy drug or combination of drugs and radiation. We review the state of the art in the fields of BMTE and HCTE and envisage dramatic transformations in stem cell and hematological cancer research and therapeutics.

## 2. Materials and Methods

A review was made by searching criteria including “Bone marrow,” “Cancers,” “Leukemia,” “Myeloma,” “Lymphoma,” “Scaffold,” “Biomimetic materials,” “Transplantation,” and “Chemotherapy” for the past 6 decades in various web-based sources such as PubMed, MeSH, Medline, Google Scholar, and others.

## 3. Results

Interest in BMTE and HCTE has been growing for the past 6 decades as evidenced by published articles (Figure 1). The use of scaffolds has accelerated since 1985, indicating the importance of BM- and HC-mimetic 3-D scaffolds in BM and HC research.

The organisms with regenerative capability regenerate the damaged tissue or whole organ on their own, without intervention of any external aid [4,5]. The advantages of 3-D culture systems replicating these features are cell–cell interactions and accurate representation of cyto-architecture for tissue regeneration. The concept of TE involves a triadic interaction of cells, scaffolds, and regulatory signals, leading to the formation of the BM or HCs tissue in a 3-D scaffold in vitro with in vivo conditions (Figure 2).

The advent of novel synthetic and biological polymer materials facilitate fabrication of 3-D architectural bioscaffold constructs with specific parameters, such as biodegradability, biocompatibility, surface morphology or porosity, and material composition, to promote cell proliferation and growth mimicking in vivo condition [6,7]. Interactions of HSC, mesenchymal stromal cell (MSC), and HC dynamic environments were studied to understand HSC maintenance and HC physiopathology in the BM environmental niche.

### 3.1. Bone Marrow Microenvironmental Niches

The bone consists of cortical bone, cancellous bone, the periosteum, the endosteum, and articular cartilage (Figure 3). The cortical bone exhibits hierarchical structural organization with osteon, Haversian canals lamella, collagen fibers, collagen fibrils, and non-collagens molecules that are in association with the collagen molecules [8,9]. It is divided into three regions, namely the endosteum, central marrow, and periosteal regions on the basis of the cell types and surrounding structural composition. The multipotent MSCs, bone-lining cells (BLC), osteoblasts, osteocytes, macrophages, and osteoclasts are involved in BM maintenance and remodeling in endosteum. The central marrow is composed of HSCs and progenitor cells (HSPC). These cells are embedded in a meshwork of structural proteins constituting the soft extracellular matrix (ECM). It provides structure and a biochemical platform for HSPC cells to interact and function [8,9,10,11]. The perivascular niche is highly vascularized fibrous connective tissue. It is a site of homing for mesenchymal cells, where they differentiate into various cell types such as adipocytes, osteocytes, chondrocytes, and endothelial and neuronal cells [12].

#### 3.1.1. Interactions of Various Cell Types to Maintain HSC Niches

Several BM cells and molecules are key players responsible for regulating the hematopoietic niche. The majority of HSCs are confined to the endosteal region and interact with a number of cells, such as MSCs, the nestin-expressing MSC population (nestin^+^ MSC), the CXCL12-abundant reticular (CAR), and macrophages that influence HSC survival and functions. It has been demonstrated that MSCs enhance the engraftment of hematopoietic stem cells in an autologous mouse transplantation model [13], and influence differentiation of HSCs [14]. Perivascular endothelial, Schwann, and sympathetic neuronal cells promote HSC maintenance and dictate HSC fate [15]. BM mononuclear phagocytes are reported to help in promoting the maintenance and retention of HSCs [16,17]. Lipid rafts, the glycoprotein microdomains, play a role in signaling processes that could enhance the responsiveness of HSPC to homing [18,19]. Several cell types and molecular interactions are involved in the maintenance of BM niches. BM homing ligands include E- and P-selectin, VCAM-1, annexin II, CXCL12, and Kit ligand. N-cadherin, annexin I, osteopontin (OPN), Ang1, BMP4, thrombopoietin, and connecxin 43 are some of the endosteal ligands. ECM interactions and intercellular adhesion lead to cell morphogenesis. The complex signaling pathways regulate HSC development from hemogenic endothelial cells (HEC) through endothelial-to-hematopoietic transition (EHT), pro-HSC and pre-HSC [20].

#### 3.1.2. Therapeutic Radiation and Chemotherapy Damage Hematopoietic Stem and Progenitor Cells (HSPC) and Recovery Strategies

Therapeutic radiation and anticancer chemotherapy drugs inadvertently damage the HSC niche and pose immense challenges for treating patients while protecting the BM niches. As radiation therapy damages HSC, MSC transplantation rescue has been in practice for the past 50 years to improve HSC recovery with the support of MSC. Radiation accelerate differentiation ability of MSC due to increased oxidative stress [21]. MSCs are resistant to radiation, which is attributed to ataxia-telangiectasia mutated (ATM) protein phosphorylation, activation of cell-cycle checkpoints, double-strand break repair, and the antioxidant capacity for scavenging reactive oxygen species (ROS) [22,23].

### 3.2. Biomimetic 3-D Scaffold for Bone Marrow and Hematological Cancer Niches

The in vitro reconstruction of the BM and HCs niches requires a thorough understanding of the natural BM and HCs microenvironments and the properties of BM and HCs mimicking materials. Bone is primarily composed of calcium hydroxyapatite with trace elements, collagen protein, and water, which provides mechanical support structure for protecting bone marrow and maintain bone and marrow homeostasis. The components of the endosteum, central marrow, and perivascular niches of the BM artificially built to mimic the structures and functions of the BM. BM cells (e.g., HSC, MSC, osteocyte, osteoblast, osteoclast etc.) in scaffold used to investigate cellular function such as mineralization and HSC maintenance. Incorporation of fat cells in bone marrow further allowed to investigate the association of bone, fat, and hematopoietic stem cells. Furthermore, inclusion of hematological cancer cells within the scaffold bone marrow, provides an important tool to test and optimize new drugs. Table 1 provides a brief summary of various aspects scaffolds that are reported earlier. Table 2 provides biomimetic 3-D in vitro bone marrow and cancer models.

### 3.3. Porosity

In the 3-D architecture of the scaffold, porosity and pore sizes play an important biophysical role. Interconnections of the pores facilitate circulation of nutrients and exchange of gases, thereby diminishing hypoxia in in vitro and in vivo conditions. The pores also support cell function with respect to attachment, migration, and proliferation. Kuboki et al. demonstrated in a rat ectopic model that pores in hydroxyapatite particles are required for osteogenesis [33]. It has been reported that the optimum pore size is 5 µm for neovascularization, 100–350 µm for regeneration of bone, and 40–100 µm for osteoid in growth. [60]. A minimum of ∼100 μm pore size is essential for cell migration and transport, and capillary formation in the bone; pore sizes >300 μm are required for enhanced new bone and capillaries formation [34]. Pore sizes generated in different fabrication methods are given in Table 3.

The pores of closed, open, blind, and through pore types can be created. The pores are created by conventional gas foaming [61,62], CO_2_-water emulsion templating [63], dense gas CO_2_ + cross-linker [64], porogen leaching [65], freeze-drying [66,67,68,69,70,71], and electrospinning methods [72,73]. As shown in Figure 4A, round pores are created in the polymer scaffold using sodium bicarbonate (NaHCO_3_) which generate CO_2_ gas in a mild acid solution [61]. In this method, the pores created are of heterogeneous sizes depending on the size of the gas bubbles emerging from the polymer slurry. In the solvent casting, casting/particle leaching method, the desired porogens are dispersed into a polymer solution and subsequently leached out by immersing the scaffold in a selective solvent, resulting in the formation of a porous network in the scaffold (Figure 4B). In this method, controlled pore sizes can be generated. Interconnected porous nonlinear filament 3-D scaffolds are fabricated by an electrospinning method of forming ECM mimicking nonlinear fibers under electro static forces (Figure 4C), and hydrogel is introduced into the stacks of nonlinear scaffolds for cell proliferation and growth [73,74]. These polymer fibers can be used for making nanocomposite scaffolds using other polymers and hydrogels.

While Table 1, Table 2 and Table 3 provide summaries of fabrication strategies for BM and cancer environment-mimicking 3-D systems, there are also major requirements for the characteristics of the scaffolds, such as mechanical strength, degradation kinetics, swelling in liquid media, and molecular linkers essential for fabrication of structural and functional stability of the scaffolds. Similarly, the HC microenvironment can be created using biomimetic synthetic polymer scaffolds, extracellular matrices (ECM), endothelial cells, and stromal cells.

### 3.4. Mechanical Sterngth and Stiffness Characterization of Bone Marrow

The mechanical properties of BM niches are varied [76] with a higher value of 435 kPa in endosteum [77], 2–10 kPa in marrow sinusoids [78,79], and 0.3 kPa in the central marrow [80]. Hydroxyapatite (HA) having higher elastic modules when fabricated with collagen forms an ECM-like scaffold mimicking the central marrow environment. The mechanical strength of the scaffold, apart from the surface chemistry and topographical features, affects MSC growth, differentiation, and regenerative capacity in the 3-D cultures [81,82,83,84]. The mineralization of BM causes enhanced stiffness. The HSCs grown in the matrix with mechanical strength of 3.7 kPa remained round and wedged in the matrix, whereas in matrix scaffolds with higher mechanical strength (>44 kPa), they are stretched and elongated, and driven to niche-mediated HSC fate decisions [85]. Human colorectal carcinoma (CRC) cell lines DLD1 and HT29 grown on 3-D laminin enriched ECM [86] and the breast cancer cell line MCF-7 grown in 3-D collagen scaffolds exhibited an enhanced expression of their respective cancer stem cell characteristics [87]. A variety of linkers and cross-linker molecules are used in the fabrication of scaffolds. They maintain the structural stability and strength of scaffolds when layers of different polymers are involved in the fabrication process. The biomimetic scaffolds are surface-modified and functioned with a variety of active biomolecules to mimic the BM and HCs environments or for any other applications. The protein chemical targets, known as primary amines (–NH_2_), carboxyls (–COOH), and sulfhydryls (–SH) account for the vast majority of cross-linking and chemical modification methods for bioconjugation. A novel phosphoramidite (PA) linker is also reported for AFM single molecule force spectroscopy experiments [88]. The cross-linker molecules establish interconnections of various biological molecules for fabrication of bioactive scaffolds.

### 3.5. Application of Biomimetic Scaffolds in Reconstion of BM and HCs Niches

In the BM environment, various cells interact and function to maintain BM homeostasis. BM niches can be created in vitro mimicking in vivo niches. Second, HCs interact with BM to create a specific BM pathophysiological environment.

#### 3.5.1. Co-Cultured Hematopoietic Stem Cells with Other BM Component Systems Modeling the BM Niche Compartments In Vitro with In Vivo Conditions

The co-culture system facilitates our understanding of the natural interaction between the cell types and its mechanisms. The knowledge of interactions helps in regulating growth and survival in the event of damages caused by irradiation and chemotherapy. HSCs in the BM interact with MSCs, macrophages, endothelial cells, and myeloma cells in cancer disease modeling of niche components in vitro. Co-cultures of HSCs and MSCs have been extensively studied [14,89,90]. Theresa Vasco’s group observed that iPSC-derived MSCs are less supportive to HSC than are primary MSCs in terms of lower long-term culture-initiating cell (LTC-IC) frequency with iPSC-MSCs as compared to primary MSCs [91]. Activated BM monocytes and macrophages preserve primitive hematopoietic cells in the bone marrow [92]. Jana Travnickova’s group demonstrated the control of HSPC mobilization and definitive hematopoiesis by macrophages [93]. The transfer of MSC-secreted ECM to MSC cultures promoted osteogenesis and bone formation in an ectopic rat model [35,36,37,94,95,96]. For in vitro 3-D model of BM tissue engineering, we used cKit enriched BM derived HSCs and demonstrated the formation of spheroids. As shown in Figure 5A,B, BM-HSC grew on PLGA and hydroxyapatite (HA) composite 3-D scaffold, forming stem cell spheroids. We co-cultured MSC and HUVEC on PLGA-HA and Matrigel composite scaffolds as a BM niche and demonstrated the formation of microvessels (MVs), as shown in Figure 5C–E.

#### 3.5.2. Biomimetic Scaffold Implantation, Not as a Prosthesis, for Desired BM Tissue Repair and Development

In tissue engineering, culture-expanded cells and scaffolds are used to produce a tissue construct for implantation. They support tissue regeneration and growth, but not as a prosthesis. In bone disorders, because of disease, irradiation, and chemotherapy, the bone environment is damaged. As a repairing strategy, scaffolds were seeded with MSCs, embryonic stem cells (ESCs), adult stem cells, induced pluripotent stem cells (iPSCs), and platelet-rich plasma (PRP) for recovering and repairing the damaged bone tissue [97]. Similarly, damaged central marrow and perivascular regions can be repaired using soft hydrogel-based scaffolds. MSCs grown on polycaprolactone (PCL) fibrous scaffolds produced paracrine products involved in tissue repair/regeneration [73]. An advanced method of biomimetic engineered bone marrow (eBM) is a “bone marrow-on-a-chip” microfluidic system. It involves culturing the cells on a scaffold in vitro, implantation in vivo, removal, and perfusion with media in microfluidic devices. It was demonstrated to have HSC and progenitor cell characteristics [40,98]. The MSC and cord blood derived HSPC cultured on hydroxyapatite coated zirconium oxide scaffold showed that HSPC remained in their primitive state (CD34^+^ CD38^−^) and were capable of forming all major colonies [53].

#### 3.5.3. Scaffold for Studying Hematological Cancers

With the advent of 3-D culture systems, malignancies, predominantly cancer cells from solid tumors such as breast cancer were utilized to understand the role of the cancer environment and cancer progression. Breast cancer cells grown in the cancer environment mimicking 3-D cultures using PLGA and PCL scaffolds showed comparative biomarker expression as observed in vivo [99]. It indicates the usefulness of scaffolds in cancer research in understanding cancer biology, anti-cancer drug screening, and control of cancer progression. Mouse CT26 colon cancer cells and BM-derived dendritic cells (BM-DC) co-cultures on an electrospun fibrous scaffold increased expression of CD86 and major histocompatibility complex Class II [100]. 3-D anisotropic collagen scaffolds were used for breast cancer cell migration studies, and showed increased migration potentials in the cancer mimicking 3-D cultures [101]. These studies indicate the substantial potential for the design and fabrication of composite scaffolds for mimicking the cancer tumor environment.

#### 3.5.4. Interaction between Hematological Cancer and Bone Marrow Niche

The HSCs reside in BM microenvironment niches that can regulate their self-renewal and differentiation. Similarly, hematological cancer stem cells (HCSCs) reside in the cancer niche, and microenvironmental cues regulate their growth and proliferation during tumor progression and development [102]. In the BM niche, MSCs interact with cancer cells, promote tumor progression and modulate the extracellular matrix (ECM) environment such that it is favorable for the invading tumor cells [103,104]. With the increasing rate of BM disorders and conditions, as well as the use of therapeutic radiation on the BM, engineered BM is considered as a potential alternative source for BM restoration. The disturbance in hematopoietic homeostasis leads to hematological cancers (HCs)/hematological malignancies and leukemic stem cell (LSC) formation. LSC/leukemia pathophysiology depends on biological cues from the BM niche for survival and proliferation. Upon administration of chemotherapeutics, LSC/leukemia cells use the BM niche as a sanctuary for survival and to escape from the therapeutic agents, acquiring chemoresistance [102,105,106,107,108,109,110]. A few recent publications are available concerning the study of hematological malignancies and their physiopathology in an ex vivo disease environment mimicking a 3-D scaffold system. The 3-D and microfluidic platform scaffolds were used to mimic the AML niche and demonstrated retention of the cells’ phenotype and proliferation, compared to the 2-D cultures [111]. Polyurethane (PU)-collagen scaffolds were used to study the biology and treatment of primary AML in an ex vivo condition [112]. Different leukemia cell lines cultured on a 3-D stromal-based model were shown to be more resistant to drug-induced apoptosis compared to effects in 2-D cultures. Similarly, N-cadherin expression in treated 3-D cultures, as compared to 2-D cultures, is indicative of cell proliferation and chemotherapy resistance [56]. Leukemia cells flourish in the BM microenvironment and may be resistant to cytotoxic drugs. Disrupting the interaction of leukemia cells and stromal cells in biomimetic polystyrene 3-scaffolds impairs their ability to gain resistance and enhances the killing effect of chemotherapy drugs [57]. Thus, BM mimetic 3-D scaffolds help in understanding the drug efficacy and toxicity studies in vitro with in vivo conditions. A poly (ethylene glycol) (starPEG)–heparin hydrogel scaffold was used to grow leukemia lines, KG1a, MOLM13, MV4-11, and OCI-AML3, and primary cells from AMLs, HUVCEs, and MSC to mimic cell interactions between AML and the vascular niche [58]. This approach is facilitating visualization of AML-vascular interactions in chemotherapeutic responses. 3-D scaffolds were used for demonstrating a mesenchymal stem cell model of the multiple myeloma (MM) bone marrow niche, indicating cytokine secretion by MSC in 3-D cultures support MM cell growth [97]. Further, they have demonstrated MSC with conserved phenotype (CD73 + CD90 + CD105+), activation of osteogenesis (MMP13, SPP1, ADAMTS4, and MGP genes), and osteoblastogenic differentiation. MSCs were grown on silk fibrous 3-D scaffolds in dexamethasone-free osteogenic media and demonstrated osteogenesis in a multiple myeloma-mimicking BM environment [59]. HSC niche mimetic 3-D scaffolds in combination with perfusion in static and dynamic cultures demonstrated the role of cytokines dose and application in vitro model for testing myeloid toxicity [55]. Further, they have demonstrated the effect of dimensionality (2-D or 3-D) and mode (static or dynamic) of HSPC/MSC co-cultures to assess myelotoxicity to 5-fluorouracil. The advanced 3-D cultured organoids are self-organized tissue architecture. They are grown from pluripotent ESCs, iPSCs, and adult stem cells [113]. They recapitulate the developmental events of tissues and organs, including natural orientation and spatial organization of different tissue specific cell types, cell–cell interactions, cell-matrix interactions, and response to biophysical cues, and more representative of vivo physiology [114]. The organoids of cancer were developed from primary colon, esophagus, pancreas, stomach, liver endometrium and emphasized the importance of organoids in cancer research [115]. The organoids grown from adult stem cells are suitable for in vitro and in vivo modeling, drug therapy, and regenerative therapy [113,116].

### 3.6. Choice of Materials and Advanced Fabrication Technologies for Scaffold Preparation

The fabrication of efficient and BM and HCs mimetic 3-D scaffolds needs to be improved with more accurate representation of the components found in the natural BM and HCs as depicted in Table 1. The basic requirements for the scaffold materials are biocompatibility, biodegradability, suitable mechanical properties, scaffold architecture, and fabrication technologies. All of these characteristics are assessed before fabricating a scaffold. Osteoconductive and time-frame biodegradability of scaffolds are critical for osteoblastic differentiation, bone regeneration, and vascularization [117,118]. The mechanical properties (Mpa) and Young’s modulus of ECM-like polymer scaffolds fabricated for various tissue engineering applications were studied [119,120,121,122,123]. The scaffold fabrication for mimicking the endosteum region needs HA with PCL in preference over a PLGA polymer because of the long degradation time for PCL, ranging from 1–2 years compared to PGA scaffolds [124].

Currently, the scaffold is prepared predominantly by manual methods. There are challenges to prepare this scaffold readily without in-depth experience. Recently introduced scaffold printing and bioprinting technologies are in the market for printing architectural and compositional elements of desired 3-D scaffolds and target tissue formations. Efforts have been made to improve accuracy, quality and reproducibility of design using computer-aided design (CAD) and fabrication of functionally graded scaffolds. In 3-D tissue printing, the target tissue elements are mixed with ECM like polymer scaffolds and printed in precise geometrical tissue or organ structures. The steps involve in selecting the organ of interest to be printed; scanning (*X*-ray, MRI or CT scan); creating graphics (Bio-CAD or Med-CAD); use of Rhinos software for modelling geometry and printing with cell types from the organ of interest [125]. Bioreactors with controlled in vivo conditions help the tissue in proper circulation of oxygen, nutrients, catabolic and anabolic waste material from the tissue. Thus, bioreactors mitigate the drawbacks of 3-D scaffold-based cell culture systems, predominantly by manual methods.

In magnetic levitation, the cells form a micro tissue rather than individual cells in the media and air interface. The micro tissue of BM-MSC formed under the magnetic field prevent spontaneous differentiation into other cell phenotypes, and retaining high expression of stem cell markers STRO-1 and nestin up to 14 days, unlike in 2-D monolayer cultures where MSC stem cell markers were lost within 3 days [126]. Magnetic levitated breast cancer (BC) and colorectal cancer (CRC) cells expressed high levels of N-cadherin and epidermal growth factor receptor molecules comparable to BC and CRC xenografts grown in severe combined immunodeficient (SCID) mice [127]. Magnetic levitated 3-D cancer cells/micro tissue was used as in vitro model for in vitro tumor development and tumor suppression therapeutics, instead of use of mice [127]. Although it is not truly representative of BM and cancer environment niches, it facilitates cell–cell interactions and aggregate growth that leads to the formation of BM and HC micro-tissue in vitro 3-D models with in vivo conditions for testing drug efficacy and toxicity. Thermoreversible injectable hydrogel-based materials are widely used for in neuro-engineering applications [74,128].

### 3.7. Future Perspectives and Conclusion

Most of the research on cancer was focused on solid cancer tumors, as such now there is a greater opportunity to focus on HCs. The potential of MSC to increase proliferation and maintenance of hematopoietic progenitor cells is harnessed in co-cultures with the HSC to enhanced engraftment of HSC in transplantation [13], formation of micro blood vessels with endothelial cells [44], multi-lineage hematopoiesis in co-cultures with differentiated osteoblasts [90] can be further exploited in 3-D co-cultures modeling the niche components in vitro using advanced scaffold systems. The role of macrophages in maintaining hematopoiesis [16,17] can be further assessed. HSC niche mimetic 3-D scaffolds may be used to study the role of cytokines and application in an in vitro model for testing myeloid toxicity. As HC cells flourish in the BM microenvironment and are resistant to cytotoxic drugs, BM and HC mimetic 3-D scaffolds help in understanding the drug efficacy and toxicity studies in vitro with in vivo conditions. The study of primary cells from AML, HUVCE and MSC to mimic cell interactions between AML and the vascular niche assists in visualization of AML-vascular interactions in chemotherapeutic responses. The MSC model of the multiple myeloma (MM) BM niche indicates cytokine secretion by MSC in 3-D cultures supporting MM cell growth. The various scaffold types used for different applications in BMTE and HCTE are summarized in Figure 6.

In view of the clinical importance of therapeutic regeneration and restoration of BM tissue after damage caused by chemo- and radiation therapy and understanding the pathophysiology of HC in BM niches perspectives and development of novel therapeutics, attention has focused on improving BM and HC mimetic 3-D scaffolds. The other technologies such as CFUs, PDMS based microfluidic systems, organ-on-chip [129,130], bone marrow-on-a-chip [110], induced pluripotent stem cells (iPSC) [131] long-term culture of human hematopoietic stem cells in a 3-D microfluidic environment [53,132] for studying specific BM niche conditions and replicating BM physiology in vitro need to be improved. Further advances in 3-D non-destructive, non-invasive tissue analysis methods help in adopting 3-D culture systems widely [60,133,134,135,136,137,138]. As the BM environment is heterogeneous in nature, a composite cell culture system with multiple cell types and advanced scaffold architecture materials are needed to replicate the various microenvironment niches and large-scale ex vivo expansion of HSC and MSC for transplantation and therapy.

The combination therapy of chemo and radiation in BM and HCs mimetic 3-D scaffolds and the scaffold-free magnetic levitation systems will open new frontiers of BM and HCs therapeutics in vitro with in vivo conditions. Biomimetic BM phantoms are used in this study for the first time as an in vitro model to study the radiation effects on hematological cancers. BM phantoms will revolutionize not only in vitro studies of hematological cancers but also radiofrequency (RF) heating and thermal monitoring studies.

Currently, such data are scanty or lacking and efforts are needed in this direction for BM phantoms and printing 3-D scaffolds and 3-D tissue.

## Figures and Tables

**Figure 1 materials-13-05609-f001:**
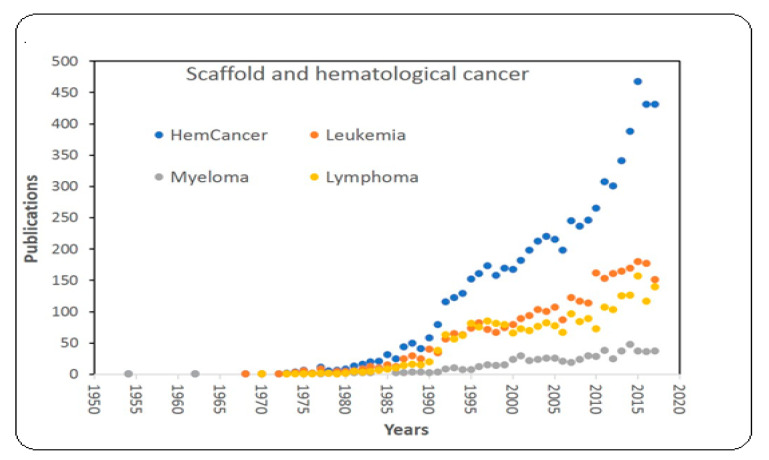
Hematological cancer data visualization: Published articles on BMTE with respect to hematological malignancies such as leukemia, lymphoma, and myeloma.

**Figure 2 materials-13-05609-f002:**
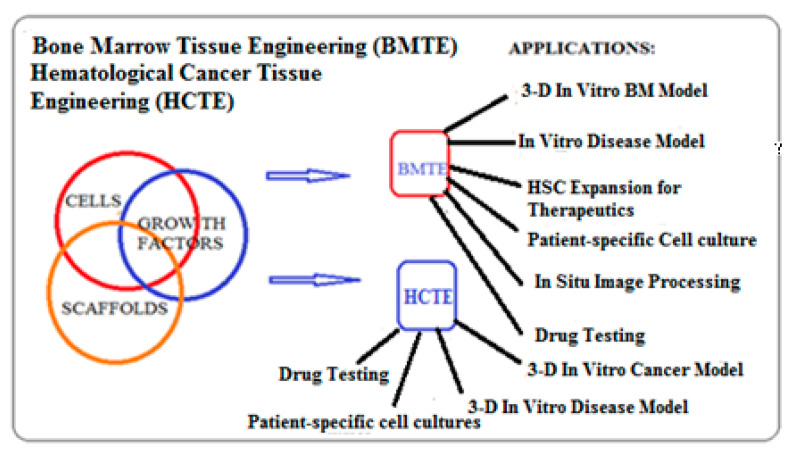
Triadic interaction of cells, growth factors, and scaffolds in BMTE and HCTE.

**Figure 3 materials-13-05609-f003:**
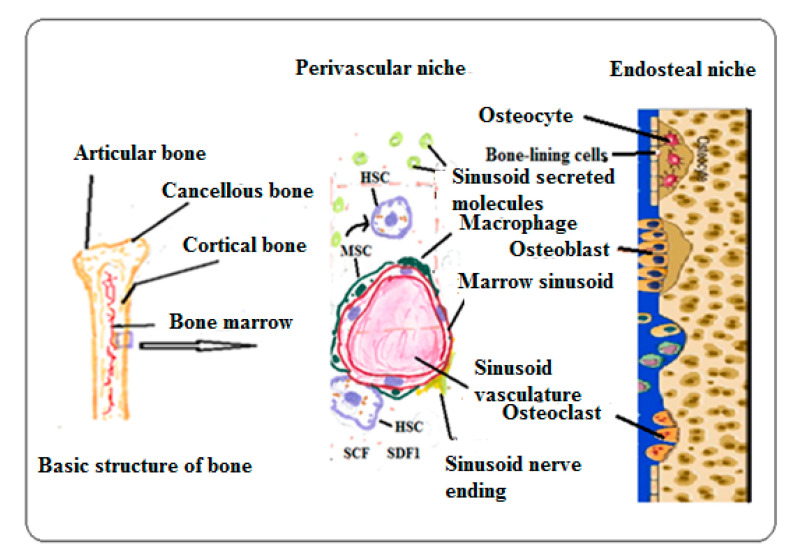
Schematic of structural composition of BM and the cellular interactions.

**Figure 4 materials-13-05609-f004:**
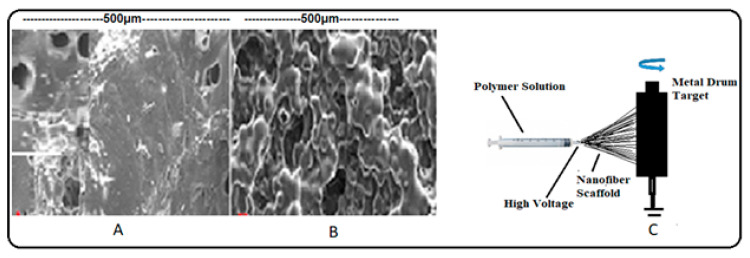
3-D scaffolds. (**A**) Pores in the PLGA scaffold generated using the gas foaming method; (**B**) Pores in the PLGA scaffold generated using solvent casting and particulate leaching method; (**C**) Schematic of electrospinning of desired polymer solution and deposition of solid fibers on the target collector. The liquid polymer transforms into solid fibers because of the effect of high voltage [73,74].

**Figure 5 materials-13-05609-f005:**
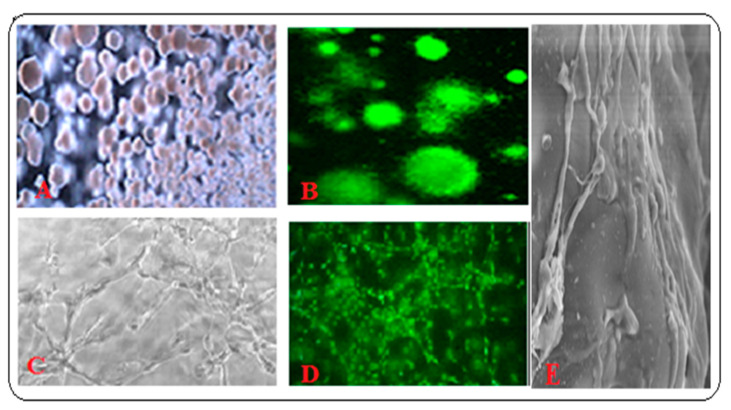
HSC from BM grown on PLGA-hydroxyapatite (HA) scaffold: (**A**) Bright field image of the spheroid colonies; (**B**) Fluorescent image after staining the colonies with acridine orange-ethidium bromide stains; (**C**,**D**) 3-D BM- MSCs and HUVEC co-cultures on PLGA-HA and Matrigel scaffold, (**E**)- SEM image of MVs. All other microscope images were taken using 5× objective.

**Figure 6 materials-13-05609-f006:**
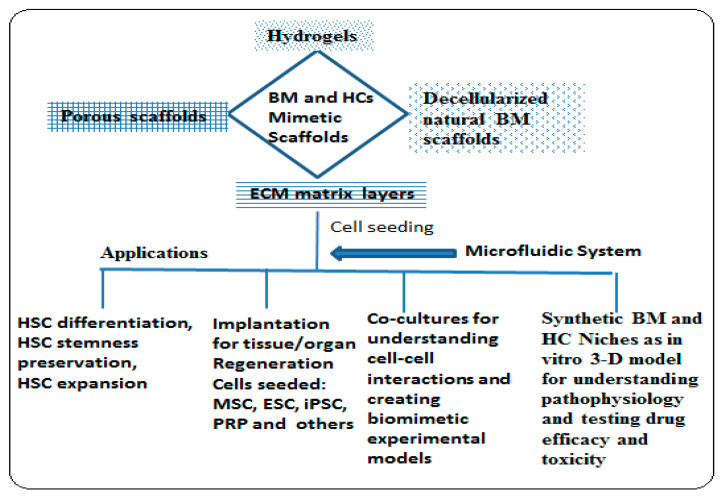
Schematic of applications of various types of scaffolds in BMTE and HCTE.

**Table 1 materials-13-05609-t001:** Comparative constituents of natural BM and synthetic scaffolds to reconstruct and replicate the BM microenvironment for transplantation and regeneration.

BM Components, Architecture and Environmental Niches	BM and HC Components Functions	Scaffolds Components Mimicking the BM and HC Microenvironment	Comments	References
**1. Bone** Bones comprise mainly collagen type I and hydroxyapatite (HA).	Creating synthetic BM and HCs niches to mimic the natural BM and HCs environments using the structural components.	Bones support the body and hold the soft organs. Marrow is a site of hematopoietic stem cells.	Convenient to fabricate artificial BM and cancer mimicking scaffolds for BM and HC studies.	[7,24,25,26]
**1a. Mineral component of bone** **1a. Calcium component** Heterogeneous composite mineral, 70% by weight of bone is a modified form of HA.	HA synthesized wet by direct precipitation of calcium and phosphate ions, and used up to 40% of HA in the scaffold fabrication. PLGA and PCL are ECM like polymers that render mechanical strength.	The hardness and rigidity of bone are due to the crystalline complex of calcium and phosphate, known as hydroxyapatite (HA).	Required amounts can be incorporated to build up structure.	[27,28]
**1b. Trace elements** Zinc, Silicon, Copper, Fluorine, Manganese, Magnesium, Iron, Boron and others elements are present.	Incorporating these trace elements into tissue engineered bone (TEB) scaffold at the time of fabrication. Doping the scaffolds with silicon, carbonate, and zinc simulate the natural bone environment.	Zinc contributes to tissue remodeling, protein and nucleic acid synthesis, cell proliferation, and remodeling ECM. Silicon is essential for bone, cartilage, organ, and connective tissues. Other elements such as copper, fluorine, manganese, magnesium, iron, and boron influence bone function.	Trace elements needed for the healthy functioning of bone cell viability and survival.	[29,30,31]
**1c. Porous** **architecture** Spongy and porous nature of the bone.	Desired pore sizes and pore microarchitecture can be created using appropriate size porogens at the time of scaffold fabrication.	Cell distribution interconnection, diffusion of nutrients and oxygen, especially in the absence of a functional vascular system.	Permeability as a function of porosity. Controlled porosity can be created in the 3-D scaffolds.	[32]
**2.Extracellular matrix** Collagen constitutes 90% of the matrix proteins, and accounts for 25 to 30% of the dry weight of bone Collagen type 1 is the predominant fraction of collagen, together with other proteins and mucopolysaccharides.	Synthesis of ECM like matrix Collagen type I and other mucopolysaccharides can be added to TEB scaffold at the time of synthesis Chitosan (CS) is another ECM-like material. It is a nontoxic, biocompatible, biodegradable cationic polysaccharide. It can be incorporated to TEB scaffold. Incorporating native or synthetic ECM into 3-D scaffolds.	Collagen with its triple helix tertiary structure and high mineralization imparts high tensile strength and high flexibility to bone. It is essential for tissue morphological organization and physiological function. Chitosan simulates the marrow environment. It also promotes electrostatic interactions with anionic glycos- aminoglycans (GAG) and proteoglycans. Incorporating native or synthetic ECM into 3-D scaffolds.	An important structural protein Chitosan is a natural biopolymer. It is easily available and widely used in tissue engineering. Direct transfer of native physio- logical and biochemical cues.	[33,34,35,36,37]
**3. BM cells** Osteoblasts, bone lining cells (BLC), osteocytes, osteoclasts, MSC, CAR cells, adipocytes, macrophages, and other cell types.	BM cells are in dynamic state of interactions with various cell types in BM environment. Studying the interactions of these different cell types help in understanding the mechanisms of their influence on HSC behavior. Varying combinations of these bone marrow cells in co-culture systems can be used for culturing in BM TE scaffold.	Osteoblasts involved in mineralization of bone and matrix proteins. Play a role in calcium homeostasis and bone resorption. Bone lining cells (BLC) function as a barrier for certain ions and induced osteogenic cells.	BM cellular functional interactions. HSC maintenance.	[16,38,39]
**4. Interaction of BM cellular components**	Co-cultures in 3-D with MSC increased proliferation and maintained HSC.	To maintain the microenvironment of hematopoietic stem and progenitor cell function.	Simulate the in vivo condition in vitro cultures.	[40,41]
**5.Blood vessels-forming cells** Interactions of multiple cell types in BM to form blood vessels.	HUVEC and MSC in perivascular niches self-assemble and form organized structures.	Blood vessel formation provides niches for hematopoietic stem cells that reside within the BM.	Vascularization facilitates the proliferation and maintenance of HSC.	[42,43,44,45]
**6. Macrophages** Macrophages are distributed in tissues throughout the body and contribute to both homeostasis and disease	Co-culture of human induced (hiPSC)—mesenchymal stem cells and macrophages recapitulate the tissue remodeling process of bone formation.	Macrophages help to retain the HC niche Through various cellular and molecular mechanisms.	HSC maintenance is performed by BM macrophages by mobilizing depleted HSC.	[16,17,46,47]
**7.BM Sympathetic nerves** They involve in BM hematopoietic homeostasis by regulating HSC maintenance genes expression. Schwann cells localize close to HSCs and maintain HSC quiescence. Chemotherapy-induced bone marrow nerve injury.	Scope for studying co-cultures of neuronal cells with HSC supporting cells. Scope for creation of nerve tissue in BM environment.	Hematopoietic stem cell hibernation in the BM niche. Involve in BM function. Adult BM cells are sources of Schwann cells	Maintain HSC quiescence Repair of impaired hematopoietic regeneration.	[48,49]
**8 Bone marrow fat** The intimate relationship among adipocytes, osteoblasts, and hematopoietic stem. Lipid rafts, the glycoprotein microdomains.	Fat components can be incorporated to the scaffolds at the time of fabrication for creating BM environment in co-culture systems.	Fat primes homing-related response of HSC/PHSC to SDF-1, through CXCR4. Fat also binds bone with calcium and forms bone grease. Play a role in signaling process, enhance the responsiveness of HSPC to homing.	The association between bone, fat, hematopoietic stem cell numbers, cytokine levels, and aging has been demonstrated.	[18,19,50,51]
**9. HSC cellular stress** Oxidative stress and hypoxia.	Study of these conditions and induced effects of radiation and cytotoxic chemotherapy in 3-D scaffold.	Understanding the damage caused by external agents to the biology of HSC.	In vitro model of HSC cellular Stress.	[52]
**10. BM niche model of tissue and fluids.**	Engineered bone marrow (eBM) on ‘bone marrow-on-a-chip’ microfluidic device is extended 3-D culture model.	Long term cultures of Bone HSC and PHSC. Myeloid toxicity studies.	Advanced stem Cell therapeutics.	[53,54,55]
**11. Hematopoietic malignancies** CLL, ALL, CML, AML, MML leukemia and multiple myeloma	Fabrication of BM and HC environments mimicking 3-D scaffolds.	BM and cancer in vitro drug testing models.	In vitro disease model.	[56,57,58,59]

As shown in Table 2, biomimetic 3-D scaffolds are fabricated for creating a microenvironment of various cellular components, including the ECM and vascular systems, with the required physical characters of native tissue.

**Table 2 materials-13-05609-t002:** Biomimetic 3-D in vitro bone marrow and cancer models.

3-D Scaffold	Materials	Methods
Solid Scaffold	PLGA, PCL, PGDA, PVA and other polymers, fats, minerals, and microelements.	Solvent casting and porogen leaching, gas foaming, freeze-drying, electrospinning, and 3-D scaffold printing.
Hydrogel	Hyaluronic acid, Chitosan, Alginate, Collagen, Gelatin, Agarose, and others	Gel casting and use of molecular cross-linkers.
Matrigel	Basement membrane extract.	Gel casting.
Biocomposite scaffold	polymers, cells, growth factors.	Bioprinting using ink-jet, laser, valve, and acoustic based.
Scaffold-free systems	No scaffold material required. Delivery of cells and active biomolecules.	Magnetic levitation and self-assembly hanging drop method for spheroid formation.

**Table 3 materials-13-05609-t003:** Pore sizes generated using various processes for different scaffolds.

Process	Polymer	Pore Size (µm)	References
Conventional gas foaming	PEGDA	100–400	[61,62]
CO_2_-water emulsion templating	Dextran	6.25–7	[63]
Dense gas CO_2_ + cross-linker	Elastin	80	[64]
Dense gas CO_2_ + cross-linker	Gelatin	80–120	[75]
Porogen leaching	PEG/PCL	180–400	[65]
Porogen leaching	PLGA	250–500	[6]
Freeze-drying	Collagen/Chitosan	50	[66]
Freeze-drying	Agarose	71–187	[67]
Freeze-drying	Chitosan, alginate	60–150	[68]
Freeze-drying	Gelatin	40–500	[69]
Freeze-drying	PVA/PCL	30–300	[70]
Freeze-drying	Chitosan/PCL	10–100	[72]
Electrospinning	Gelatin/PCL	20–80	[72,74]

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
