# Peer review of "3-D Cell Culture Systems in Bone Marrow Tissue and Organoid Engineering, and BM Phantoms as In Vitro Models of Hematological Cancer Therapeutics—A Review"

_materials, 2020, doi:10.3390/ma13245609_

Round 1

Reviewer 1 Report

The authors provided a review about the generation and use of scaffolds and 3D systems as in vitro models to study the BM and HC microenvironments in pathological conditions. I have the following comments:

  • The authors focused on engineered BM tissue as in vitro models of haematological cancer therapeutics, but then they describe only the BM niches in physiological conditions. I suggest adding a new section about interaction between haematological cancer cells and their niche, also showing the differences with respect to the healthy niche microenvironment. I also suggest moving the text from line 31 to 36 (appropriately reformulated) from the introduction to this new section about interaction between cancer cells and BM niche. Thus, obviously reformulated also the introduction.

  • I suggest reorganizing the table 1 giving the following order to the columns: BM components, architecture and environmental niches; BM and HC components functions; scaffolds components mimicking the BM and HC microenvironment; Comments; References.

  • In table 1, in the first row, the authors reported that "synthetic BM and HC niches can be created in vitro using structural components", can the authors give an example to a better clarification?

  • In table 1, in the row about extracellular matrix, please adjust the layout of all the paragraphs about chitosan. In addition, I suggest including the information contained into the row about “in vitro source of ECM” into the previous row (extracellular matrix), since the topic is very similar

  • Please reformulate in a more detailed manner the sentence “BM cellular functional interactions” and “HSC maintenance” in the last column of table 3, since now it is not very clear.

  • I think that the information contained in all the row about sympathetic nerve is not clear, please reformulate. In addition, the sentence “repair of impaired hematopoietic regeneration” appear twice, please correct.

  • Please, add the scale bar in figure 5.

  • At line 307, the authors stated that 3D culture organoids are obtained from pluripotent ESCs, iPSCs and adult stem cells, can you provide a reference?

Author Response

We would like to thank the reviewers for their time and useful comments. We believe we have suitably responded to their concerns and provide a ‘point-by-point’ response to their comments. Our response is presented below in red.

Reviewer 1

  • The authors focused on engineered BM tissue as in vitro models of haematological cancer therapeutics, but then they describe only the BM niches in physiological conditions. I suggest adding a new section about interaction between haematological cancer cells and their niche, also showing the differences with respect to the healthy niche microenvironment. I also suggest moving the text from line 31 to 36 (appropriately reformulated) from the introduction to this new section about interaction between cancer cells and BM niche. Thus, obviously reformulated also the introduction. 
  • Response: As suggested, we created new section 3.5.4 Interaction between hematological cancer cells and their niche. The text from line 31 to 36 (appropriately reformulated) from the introduction deleted and moved to this new section 3.5.4
  • I suggest reorganizing the table 1 giving the following order to the columns:  
  • Response: As suggested, table is rearranged in following orders: BM components, architecture and environmental niches; BM and HC components functions; scaffolds components mimicking the BM and HC microenvironment; Comments; References.
  • In table 1, in the first row, the authors reported that "synthetic BM and HC niches can be created in vitro using structural components", can the authors give an example to a better clarification? 
  • Response: We regret including a sentence about our development and we removed the specific sentence. References and review were given for previously published work. While we are in the process of developing BM phantoms using synthetic bone and marrow components, this will require a thorough material characterizations and testing biological application, which is beyond the scope of this publication. We hope, once the study is complete, we can report it in near future as a separate communication.
  • In table 1, in the row about extracellular matrix, please adjust the layout of all the paragraphs about chitosan. In addition, I suggest including the information contained into the row about “in vitro source of ECM” into the previous row (extracellular matrix), since the topic is very similar 
  • Response: Adjusted as suggested
  • Please reformulate in a more detailed manner the sentence “BM cellular functional interactions” and “HSC maintenance” in the last column of table 3, since now it is not very clear. 
  • Response: This is in table 1. We rewrote in the table 1 (item: 3. BM cells) that BM cells are in dynamic state of interactions with various cell types in BM environment. Additional relevant references were added.
  • I think that the information contained in all the row about sympathetic nerve is not clear, please reformulate. In addition, the sentence “repair of impaired hematopoietic regeneration” appear twice, please correct. 
  • Response: We removed the repeat. Sympathetic nerve involves in BM hematopoietic homeostasis by Regulating HSC maintenance genes expression. Adult BM cells are sources of Schwann cells.
  • Please, add the scale bar in figure 5.      
  • Response: Added scale bar and magnification
  • At line 307, the authors stated that 3D culture organoids are obtained from pluripotent ESCs, iPSCs and adult stem cells, can you provide a reference?
  • Response: We have added a reference [Jihoon Kimet al. Human organoids: model systems for human biology and medicine. Nature Reviews Molecular Cell Biology (2020)].

Reviewer 2 Report

Taking into account all the several features, the accuracy, scientific quality, scientific content and the interpretation of the results are very good.

- The approach is interesting and the topic is appropriate for the journal.

  • The work has a very clear structure and all the sections are well written in a way that is easy to read and understand. In addition, the structure of the paper is very good.

  • The paper deals with a review on 3-D Cell Culture Systems in Bone Marrow Tissue and Organoid Engineering, and BM 4 Phantoms as In Vitro Models of Hematological Cancer Therapeutics, reporting interesting results. In the introduction, the authors start to discuss about cells, cell differentiation, bone and marrow tissue engineering, 3D scaffolds/platforms and biomimetic structures. Accordingly, I suggest to better stress and BRIEFLY cite some concepts and progresses related to the design of 3D hierarchically controlled structures/platforms (generally obtained through additive manufacturing) and functional hydrogel/gel-based constructs also employed for several applications (i.e., “Systematic Analysis of Injectable Materials and 3D Rapid Prototyped Magnetic Scaffolds: From CNS Applications to Soft and Hard Tissue Repair/Regeneration”. Procedia Eng 2013; 59: 233–239.…) evidencing some features which should play a crucial role in the field. Then, the authors should continue to report their study on the features related to 3-D Cell Culture Systems in Bone Marrow Tissue and Organoid Engineering, and BM 4 Phantoms as In Vitro Models of Hematological Cancer Therapeutics. All of this should improve the quality of the paper, reporting the progresses in developing design strategies and analyses related to 3D hierarchically controlled structures/platforms and functional hydrogel/gel-based constructs, thus helping the different kinds of readers to better understand the value of their work.    
  • The introduction should be improved.
  • The List of references should be improved.
  • It seems that the paper does not contain repetitions.
  • The quality of some figures should be improved.
  • The title is adequate and appropriate for the content of the article.
  • The abstract contains information of the article.
  • Figures and captions are essential and clearly reported.

Author Response

We would like to thank the reviewers for their time and useful comments. We believe we have suitably responded to their concerns and provide a ‘point-by-point’ response to their comments. Our response is presented below in red.

Reviewer 2

Comments and Suggestions for Authors

Taking into account all the several features, the accuracy, scientific quality, scientific content and the interpretation of the results are very good. - The approach is interesting and the topic is appropriate for the journal. The work has a very clear structure and all the sections are well written in a way that is easy to read and understand. In addition, the structure of the paper is very good.

Response: We appreciate reviewer’s encouraging comments.

The paper deals with a review on 3-D Cell Culture Systems in Bone Marrow Tissue and Organoid Engineering, and BM 4 Phantoms as In Vitro Models of Hematological Cancer Therapeutics, reporting interesting results. In the introduction, the authors start to discuss about cells, cell differentiation, bone and marrow tissue engineering, 3D scaffolds/platforms and biomimetic structures. Accordingly, I suggest to better stress and BRIEFLY cite some concepts and progresses related to the design of 3D hierarchically controlled structures/platforms (generally obtained through additive manufacturing) and functional hydrogel/gel-based constructs also employed for several applications (i.e., “Systematic Analysis of Injectable Materials and 3D Rapid Prototyped Magnetic Scaffolds: From CNS Applications to Soft and Hard Tissue Repair/Regeneration”. Procedia Eng 2013; 59: 233–239.…)

Response: Thank you. We have expanded this paragraph and included suggested reference.

evidencing some features which should play a crucial role in the field. Then, the authors should continue to report their study on the features related to 3-D Cell Culture Systems in Bone Marrow Tissue and Organoid Engineering, and BM 4 Phantoms as In Vitro Models of Hematological Cancer Therapeutics. All of this should improve the quality of the paper, reporting the progresses in developing design strategies and analyses related to 3D hierarchically controlled structures/platforms and functional hydrogel/gel-based constructs, thus helping the different kinds of readers to better understand the value of their work.    

Response: With due respect, as this paper is to review published work of scaffold system, and our laboratory based scaffold development is at an early stage. Reporting of new scaffold development will require detail, characterization, validation and specific application will require full report, beyond the scope of the current review manuscript. However, we will be working hard to validate the system and communicate later.

Reviewer 3 Report

  • Do you have any idea or evidence for the scaffolds which made from the inorganic materials such as calcium phosphates (hydroxyapatite)?
  • In Table 3, you mentioned pore sizes for different scaffolds. However, You described the phenomena after in vivo experiment with PLGA/HA scaffolds in Fig. 5. I think the bioreaction for composites (PLGA/HA) in vivo environment could be effected by the content of inorganic materials in composites. Therefore, it would be better to add the description about the scaffolds including the inorganic materials (hydroxyapatite) and the composites (polymer/inorganic materials).
  • As you know, the scaffolds should have the appropriate pore size and mechanical properties. In this paper, there is not much the explanation of the mechanical properties. If you have the references or stuffs about the mechanical properties for the sacffolds, Could you add in paper ?

Author Response

We would like to thank the reviewers for their time and useful comments. We believe we have suitably responded to their concerns and provide a ‘point-by-point’ response to their comments. Our response is presented below in red.

Reviewer 3

Comments and Suggestions for Authors

Language and organization need to be improved. Article is very confusing and hard to read. Couple of spelling errors throughout article. 

Response: We have revised the manuscript with corrections.

Section 3.2 needs to be expanded; it contains only 3 sentences. Having tables does not mean no discussion is needed. 

Response: As suggested, we have expanded the paragraph.

Please be careful about terms used e.g. mechanical strength is not the same as mechanical stiffness. 

Response: Thank you. We have corrected this. We agree that mineralization of BM causes enhanced stiffness.

Section 3.5.4 seems out of place under "Application of biomimetic scaffolds...".

Response: Section 3.5.4 is renamed as 3.6

Reviewer 4 Report

Language and organization need to be improved. Article is very confusing and hard to read. Couple of spelling errors throughout article. 

Section 3.2 needs to be expanded; it contains only 3 sentences. Having tables does not mean no discussion is needed. 

Please be careful about terms used e.g. mechanical strength is not the same as mechanical stiffness. 

Section 3.5.4 seems out of place under "Application of biomimetic scaffolds...".

Author Response

None

Round 2

Reviewer 1 Report

The authors performed all the requested revisions

Reviewer 4 Report

Manuscript reads better after revision.